# Understanding the relationship between apathy, cognition and functional outcome in schizophrenia: The significance of an ecological assessment

Daniela Ramos-Mastache[1,2], Alejandra Mondragón-Maya[2]*, Edith J. Liemburg[3], Stefanie Enriquez-Geppert[4], Katharina S. Goerlich[1], Mauricio Rosel-Vales[5], David Pérez-Ferrara[2], Ashok S. Jansari[6], Andre Aleman[1,3,7]

**1** Cognitive Neuroscience Center, Department of Biomedical Sciences of Cells and Systems, University Medical Center Groningen, Groningen, Netherlands, **2** Facultad de Estudios Superiores Iztacala, Universidad Nacional Autónoma de México, Tlanepantla de Baz, Estado de México, México, **3** Rob Giel Research Center, Department of Psychiatry, University Medical Center Groningen, Groningen, Netherlands, **4** Department of Developmental and Clinical Neuropsychology, University of Groningen, Groningen, the Netherlands, **5** Clínica de Esquizofrenia, Instituto Nacional de Psiquiatría Ramón de la Fuente Muñiz, Ciudad de México, México, **6** Department of Psychology, Goldsmiths, University of London, London, United Kingdom, **7** Shenzhen Key Laboratory of Affective and Social Neuroscience, Center for Brain Disorders and Cognitive Sciences, Shenzhen University, Shenzhen, China

* ale.mondragon@comunidad.unam.mx

## Abstract

In recent years there has been an increasing interest in understanding the role apathy plays in mediating the relationship between cognitive impairment and functional outcome. In general, most studies measure cognition with traditional cognitive tests that give explicit instructions and guide the participants toward generating a response. However, given that apathy is defined by a decrease in self-initiated behavior, it is crucial to evaluate cognition with ecological tasks that do not explicitly direct the patient´s motivation to generate behaviors to assess the actual effect. This study investigated whether an ecological cognitive assessment (the Jansari Executive Function Assessment, JEF[©]) would uniquely contribute to the relationship between cognition, apathy, and functional outcome in schizophrenia. The Apathy Evaluation Scale (AES), neuropsychological tests and the JEF[©] were administered to 20 patients with schizophrenia. Hierarchical multiple regression and mediation analysis were performed to test the associations between the variables of interest. Results showed that JEF[©] explained a significant portion of the variance in AES (25%). In addition, apathy explained 36% of the variance in functional outcome. However, AES did not mediate between cognition and functional outcome. Our results highlight the importance of assessing cognition with tasks that require integration of cognitive functions needed for real life demands.

**Data Availability Statement:** The data has been uploaded to https://figshare.com DOI assigned to the database is: 10.6084/m9.figshare.21158947.

**Funding:** The authors received no specific funding for this work.

**Competing interests:** The authors have declared that no competing interests exist.

# Introduction

One of the most critical aspects in patients with schizophrenia is the impairment they exhibit in functional outcome [1], which can be understood as the set of daily activities that manifest in areas such as occupational functioning, psychosocial functioning, cognition and leisure. Due to these dysfunctions, people who suffer from schizophrenia can exhibit difficulties in independent living. This highlights that the ultimate goal of schizophrenia research should be to find the determinants of functional outcome that can be treated.

There is evidence suggesting that cognitive impairment and motivational deficits are predictors of functional outcome in schizophrenia patients. Indeed, cognition explains between 20–60% of the variance in patients' functional outcome [2], and motivational deficits may exert an even stronger effect [3, 4]. Nakagami et al. [5] explored the relationship between motivational deficits, cognition, and psychosocial functioning in a group of 120 patients with schizophrenia. After including motivational deficits as a mediator variable into a mediation analysis model, the direct pathway between cognition and psychosocial functioning lost its initial significance, suggesting that cognition influences functioning thru motivation. Similarly, a study by Liemburg et al. [6] suggested amotivation [a subdomain of negative symptoms comprised by items from the Positive and Negative Syndrome Scale (PANSS)] to have a mediating role between cognition and functional outcome in schizophrenia.

In the context of schizophrenia, the measurement of motivational deficits is embedded in clinical scales that assess the patient's positive and negative symptoms [Brief Negative Symptom Scale [7]; Scale for the Assessment of Negative Symptoms [8]; Positive and Negative Syndrome Scale [9]; Brief Psychiatric Rating Scale [10]]. These instruments provide information on motivation, although limited because these scales were not primarily developed for measuring problems of motivated goal-directed behavior. The Apathy Evaluation Scale (AES) [11], could provide a better way to assess the mediating role of motivational deficits in patients with schizophrenia. This scale is a well-established measure of apathy that has already been used in schizophrenia and psychosis research [12–15], albeit not as a putative mediator between cognitive performance and measures of functional outcome. Apathy, one of the core negative symptoms of schizophrenia, is conceptualized as a motivational impairment characterized by observable changes in goal-directed cognition, emotion, and behavior [16]. Its definition implies that patients show apathy when they express a reduction in activity due to a lack of self-initiated motivation for goal-directed behavior and that this reduction is not attributed to a decrease in consciousness, cognitive impairment, or emotional disturbances [17]. Given that apathy is defined by a decrease in voluntary and self-initiated behavior, it would be sensible to evaluate cognition using tasks that do not explicitly direct the patient´s motivation to generate behaviors.

Importantly, traditional neuropsychological tests that are used to assess cognition in schizophrenia research may have little ecological validity when it comes to assessing the relationship between cognition, motivation, and everyday functioning. This is because traditional tests are relatively limited in scope since they mostly assess an isolated cognitive function primarily on artificial settings [18]. However, in real life settings, cognitive task-driven motivational processes require multiple integrated functions [19]. A measure with a higher ecological validity is the Jansari Assessment of Executive Functions (JEF©) [20], a task embedded in a virtual office environment. This assessment encourages the participant to generate creative solutions without relying on direct instructions to guide their behavior. So far, this task has not been used in schizophrenia research, but has been shown to be a valid and reliable test for executive functioning (EF) in patients with acquired brain injury [20], frontal lobe lesions [21] and patients with affective disorders [22]. Notably, the latter study found that JEF© scores predicted

performance on a global cognition composite measure based on neuropsychological tests and a performance-based measure of functional capacity.

The aim of the present study was twofold: First, to test whether a cognitive ecological assessment would be a more valid approach to understanding the relationship between cognition, apathy, and functional outcome in schizophrenia. The second aim was to test whether apathy levels mediate between the JEF© and functional outcome. We hypothesized that the ecological assessment would have a stronger correlation with apathy than a composite score of EF cognitive tests. Further, we predicted that apathy would mediate between neurocognition and functional outcome.

## Method

### Study population

The study was conducted at the National Institute of Psychiatry "Ramón De La Fuente Muñiz" in Mexico (INPRFM). The study was approved by the ethics committee of the INPRFM which are in accordance with the Declaration of Helsinki (register number: SC18086.0) and the participants gave written informed consent prior to participation. Patients who had been diagnosed with schizophrenia at least one year before the study by specialized psychiatrists based on the DSM-5, aged between 20 and 55 years were invited to participate. Initially, a total of 63 participants were to be recruited for the study, however due to Covid-related restrictions, only 35 participants could be included. Of these, 11 participants (8 women and 3 men) withdrew their participation after the sociodemographic data collection and four patients (three women, one man) could not complete the assessment. Thus, the sample consisted of 20 outpatients with schizophrenia (3 women and 17 men) from the INPRFM. Exclusion criteria were additional neurological disorders, substance dependence (except for nicotine), severe hearing or vision problems and severe catatonic symptoms.

### Measurements

**Assessment of cognitive functions.**   The ecological evaluation of EF was defined by the overall score obtained from the JEF© in its Spanish version [20]. This computerized task takes place in a virtual environment in which the participant works as an assistant in an office. During the assessment the participant must solve problems that may arise in a real work office, such as sorting a list of tasks from most to least important or decide what to do when a water leak cannot be fixed. This task consists of 17 items divided into eight constructs assessing different executive behaviors qualitatively and quantitatively (planning, prioritization, selection, creative, adaptiveness, action-based prospective memory, event-based prospective memory, and time-based prospective memory). Each item can achieve the value of "2" if it is completely fulfilled, "1" if it is done incompletely and "0" if the task is not done at all. The sum of the 17 items results in an overall score of the participant's performance.

Additionally, EF were assessed with the Brief Assessment of Cognition in Schizophrenia Symbol Coding (BACS-SC), a time-based paper and pencil task in which the participant uses a key to write digits that correspond to specific symbols [23]; Trail Making Test A and B, a time-based paper and pencil tasks in which the participant draws a continuous line that orders numbers or letters that are placed disorderly on a sheet of paper; and Verbal Fluency Test, in which the participant is asked to enumerate as many animal names as possible within one minute [24]. Scores of these tests were converted into z-scores and adjusted so that lower scores were indicative of worse cognitive performance. These scores were later combined to obtain a composite score of EF measures (CEF; S1 Table). Finally, the vocabulary subscale from the WAIS-III was administered to ensure that the participants have the verbal level to comprehend

the JEF© instructions. All participants with a scalar score under eight were withdrawn from the study, all participants in the study met this criterion.

**Apathy measure.**   The degree of apathy was evaluated using the Apathy Evaluation Scale [11] in its clinical version (AES-C). A Spanish version of the AES-C was adapted using the translation-backtranslation method [25]. The manual and the score sheet were translated into Spanish by native speaking psychologists fluent in English, later this translated version was backtranslated into English by other Spanish speaking psychologist with fluent English speaking skills. Finally, the two versions were compared by a committee formed by all the involved translators until conceptual, semantic, and content equivalence were achieved. The AES-C consists of 18 items that evaluate three components of apathy: cognition, behavior, and emotion. The items are rated according to a four-point Likert-type scale ranging from "not at all" (1), "slightly", "somewhat" to "a lot" (4). The higher the score the higher the levels of apathy.

**Daily life functioning measure.**   Participants' functionality in daily life was assessed with the Functioning Assessment Short Test (FAST) in its Spanish version [26]. This is an interview scale that consists of 24 items which evaluate autonomy, occupational functioning, cognitive functioning, financial issues, interpersonal relationships, and leisure time. Scores for each item range from zero to four points; by their sum a total score for the overall performance of the participant is obtained, where higher scores are indicative of more impairment in daily life functioning.

**Clinical measures.**   Furthermore, different clinical evaluations were performed. Symptoms of depression, substance abuse and severity of positive and negative symptoms were assessed with the Calgary Depression Scale (CDS) [27], the Mini-International Neuropsychiatric Interview (MINI) Spanish version 5.0.0 [28], the Brief Negative Symptom Scale (BNSS) [7], and the five-dimension version of the Positive and Negative Schizophrenia Syndrome (PANSS) scale [9].

## Procedure

Patients were invited by their psychiatrist at the INPRFM. After patients read and signed the informed consent, an interview was performed to verify compliance with the inclusion criteria. Subsequently, clinical and neuropsychological measures were applied, followed by the EF assessment. The duration of the evaluation was approximately three hours. Breaks were given to the participants between assessments if required. The EF assessment was conducted by specialized neuropsychologists while the clinical evaluations were performed both by psychiatrists and psychologists.

## Statistical analyses

To exclude highly correlated variables from the subsequent regression analysis bivariate Pearson correlations between the overall scores of cognition (i.e., JEF© and CEF), apathy (AES-C), functionality, clinical measures (i.e., FAST, PANSS, CDS, BNSS) and sociodemographic characteristics (age, years of education and age of onset of illness) were performed. Partial correlation analyses were conducted to assess whether changes occurred in the relationship between the AES-C, FAST and cognitive test performance when controlling for depression (CDS). Additionally, Spearman correlations were performed between cognitive scores, apathy, clinical measures, gender, and type of medication to discard possible confounding variables.

To test whether a cognitive ecological assessment would be a more valid approach to understanding the relationship between cognition, apathy, and functional outcome in schizophrenia, a first hierarchical multiple regression was conducted with AES-C as the dependent variable. The JEF© was introduced as the first predictor and CEF as the second predictor, to test whether

CEF would explain additional variance independently of JEF©. In addition, to test whether AES-C would explain additional variance on FAST independently of JEF©, a second hierarchical multiple regression was conducted, with FAST as the dependent variable. The JEF© was introduced as the first predictor and the AES-C as the second predictor.

Finally, to test whether apathy mediated between neurocognition and functional outcome, a mediation analysis was performed using the PROCESS V2.16 toolbox [29], where the overall score of the JEF© was used as a predictor variable, the AES-C total score as a mediator variable, and the FAST overall score as outcome measure. All statistical analysis were performed using the IBM SPSS statistic version 22.0. An overview of all the variables can be found in S1 Table.

## Results

Means and standard deviations of the participants'sociodemographic information, clinical measures and cognition scores can be found in Table 1. Normal distribution was confirmed with the Shapiro Wilk test. Results of Pearson correlations between cognitive measures, clinical measures and sociodemographic showed that years of education correlated with AES-C (r = -.661, p < 0.01), JEF© (r = .683, p < 0.01), FAST (r = -.624, p < 0.01), BNSS (r = .800, p < 0.01) and PANSS (r = -.606, p < 0.01) (S2 Table). Further, Spearman correlations between apathy, functionality, cognitive measures and sociodemographic showed that benzodiazepines (r = .481, p = .032) and antidepressants (r = .490, p = .028), correlated with CDSS and antipsychotic medication with BNSS (negative symptoms; r = .509, p = .022).

Results of bivariate correlations among cognition assessments and clinical measures of apathy, functional outcome, depression, and symptom severity are shown in Table 2. Ecological and composite scores of EF tests correlated moderately with each other. Specifically, JEF©

**Table 1. Sociodemographic and clinical data from the sample.**

| Characteristics | | N | % |
|---|---|---|---|
| **Gender** | Female | 3 | 15 |
| | Male | 17 | 85 |
| **Antipsychotic medication** | Typical | 2 | 10 |
| | Atypical | 17 | 85 |
| | Both | 1 | 5 |
| **Other pharmacological treatment** | Benzodiazepines | 5 | 25 |
| | Antidepressants | 9 | 45 |
| | | **Mean** | **SD** |
| **Age in years** | | 37.15 | 11.14 |
| **Years of education** | | 13.20 | 3.38 |
| **Age of onset of illness** | | 24.20 | 5.88 |
| **AES-C** | | 46.55 | 11.05 |
| **JEF** | | 13.20 | 6.25 |
| **CEF** | | -0.88 | 2.45 |
| **FAST** | | 31.15 | 14.10 |
| **BNSS** | | 33.55 | 16.11 |
| **CDS** | | 3.00 | 2.97 |
| **PANSS** | | 72.65 | 15.07 |
| **WAIS-III. Vocabulary** | | 10.53 | 2.70 |

AES-C = Apathy Evaluation Scale—Clinical version; JEF© = Jansari Assessment of Executive Functions; CEF = Composite score of executive functioning; FAST = Functionality Assessment Short Test; BNSS = Brief Negative Symptom Scale; CDS = Calgary Depression Scale; PANSS = Positive and Negative Syndrome Scale.

**Table 2. Correlations among measures of schizophrenia patients.**

| | 1 | 2 | 3 | 4 | 5 | 6 | 7 | 8 | 9 | 10 | 11 | 12 |
|---|---|---|---|---|---|---|---|---|---|---|---|---|
| **1. AES-C** | 1.000 | | | | | | | | | | | |
| **2. JEF©** | -.546* | 1.000 | | | | | | | | | | |
| **3. CEF** | -.380 | .457* | 1.000 | | | | | | | | | |
| **4. FAST** | .817** | -.567** | -.474* | 1.000 | | | | | | | | |
| **5. BNSS** | .680** | -.636** | -.278 | .503* | 1.000 | | | | | | | |
| **6. CDS** | .083 | -.088 | .008 | .287 | .037 | 1.000 | | | | | | |
| **7. PANSS Positive** | -.087 | -.410 | -.117 | -.031 | .288 | .279 | 1.000 | | | | | |
| **8. PANSS Negative** | .501* | -.291 | -.185 | .389 | .739** | -.084 | .150 | 1.000 | | | | |
| **9. PANSS Cognitive / Disorganization** | .350 | -.487* | -.349 | .526* | .437 | .135 | .315 | .537* | 1.000 | | | |
| **10. PANSS Excitability / Hostility** | .013 | -.212 | -.163 | .205 | -.257 | -.331 | -.172 | -.207 | .105 | 1.000 | | |
| **11. PANSS Depression / Anxiety** | .142 | -.193 | -.316 | .162 | .108 | .366 | .073 | .030 | .209 | -.005 | 1.000 | |
| **12. PANSS Total** | .423 | -.698** | -.440 | .506* | .565** | .159 | .485* | .577** | .842** | -197 | .439 | 1.000 |

AES-C = Apathy Evaluation Scale—Clinical version; JEF© = Jansari Assessment of Executive Functions; CEF = Composite score of executive function assessment;

FAST = Functionality Assessment Short Test; BNSS = Brief Negative Symptom Scale; CDS = Calgary Depression Scale; PANSS = Positive and Negative Syndrome Scale.

* significance level at .05

** significance level at .01.

correlated significantly with the AES-C and FAST, while CEF correlated only with FAST. This analysis also showed that the measures of negative symptoms (PANSS negative and BNSS) correlated with the apathy measure and that PANSS cognitive and PANSS total score correlated with JEF©. Notably, the CEF did not correlate with any negative symptom measure. Depression did not alter the correlation coefficients between apathy, cognitive measurements, and functional outcome (Table 3).

In the first step of the first hierarchical multiple regression, JEF© scores were entered, explaining a substantial portion of the variance in AES-C (25.9%), (F(1,19) = 7.657, p = .013). Adding CEF in the second step added 2.1% of the explained variance, a non-significant increase, $F_{change}$ = .535, p = .475. In the first step of the second hierarchical multiple regression, JEF© scores were entered, explaining a substantial portion of the FAST variance (28.4%), (F(1,19) = 8.548, p = .009). However, adding AES-C in the second step resulted in an additional increase of explained variance in FAST [($R_{change}$ = 36.7%), $F_{change}$ = 20.045, p < .001]. Finally, results of the mediation analysis are shown in Fig 1 and Table 4. The univariate regression analysis revealed significant associations between cognition and apathy (X→M) and between apathy and functional outcome (M→Y). However, apathy did not show a mediating effect between cognition and functional outcome.

**Table 3. Partial correlations between apathy, cognition and functional outcome when controlling for depression.**

| Control Variable | | AES-C | JEF | CEF | FAST |
|---|---|---|---|---|---|
| **CDS** | AES-C | 1.000 | | | |
| | JEF© | -.543* | 1.000 | | |
| | CEF | -.382 | .459* | 1.000 | |
| | FAST | .831** | -.568* | -.497* | 1.000 |

AES-C = Apathy Evaluation Scale—Clinical version; JEF© = Jansari assessment of Executive Functions; CEF = Composite score of executive function assessment;

FAST = Functionality Assessment Short Test; CDS = Calgary Depression Scale.

* significance level at .05

** significance level at .01.

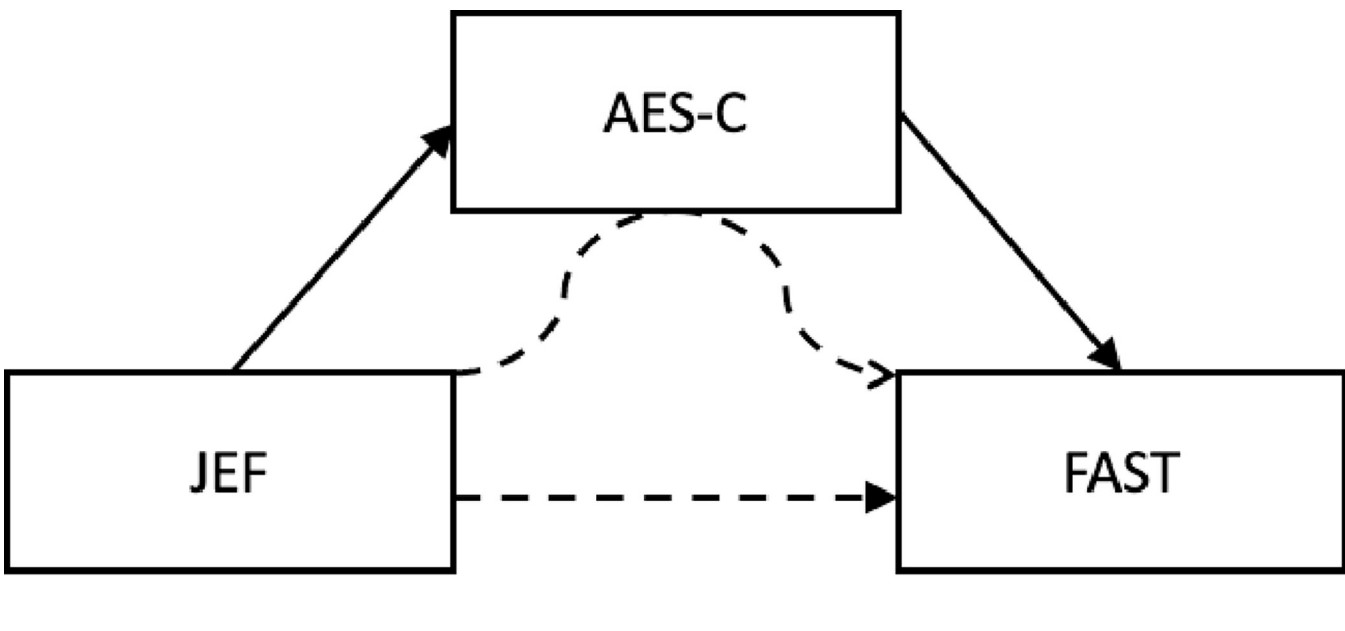

**Fig 1. Cross-sectional mediation pathway between the predictor (cognition), mediator (apathy) and outcome (functional outcome).** Black arrows indicate significant association while dotted arrows indicate no significant association. The outer straight arrows indicate direct effects, the curved inner arrow the indirect, mediating effect.

## Discussion

The first aim of the present study was to investigate whether an ecological cognitive assessment of EF (with the JEF©) would show a stronger association with apathy and functional outcome in comparison to a composite score of an EF assessment (CEF) comprising standard neuropsychological tests. The second aim was to find out whether a validated measure of apathy mediated the relationship between ecologically assessed EF and functional outcome in patients with schizophrenia.

Hierarchical multiple regression analysis showed that the JEF© explained a substantial portion of the variance of AES-C (25.9%), in contrast to the CEF. Additionally, the JEF© showed a

**Table 4. Results of the mediation analysis.**

| Association | Coeff. | SE | t | P | CI |
|---|---|---|---|---|---|
| X → M | -0.97 | 0.35 | -2.77 | 0.013 | -1.70 – -0.23 |
| M → Y | 0.92 | 0.21 | 4.48 | 0.0003 | 0.49–1.36 |
| X → Y | -0.39 | 0.36 | -1.07 | 0.30 | -1.16–0.38 |
| X → M → Y | -0.89 | 0.60 | - | - | -2.46–0.11 |

Coeff. = regression coefficient; SE = standard error; t = t test statistic; P = p-value; CI = confidence interval; Predictor X = JEF overall; Mediator M = AES-C total; Outcome Y = FAST total.

stronger correlation with AES-C and with other scales that measure negative symptoms (BNSS and PANSS), compared to the CEF. The computer based virtual environment of the JEF[©] assessment allowed the participants to respond in a more similar way compared to real life daily situations. The CEF, in contrast, is characterized by simpler (i.e., less multidimensional) tests with a defined and limited task instruction that might guide the participant to generate a response. It thus seems that in the case of schizophrenia, the JEF[©] takes into account other aspects such as initiating behavior and integration of attention and executive performance. Consistent with our results, it has been suggested that construct tests are aimed to assess a specific cognitive domain, whilst the ecological or functioning tests are more sensitive to the integration of domains involved in the cognitive pathology of schizophrenia patients [22]. Fervaha et al. [30] proposed that the relationship between motivation and cognition may be the result of deficits in the lack of mental effort required to perform a cognitive activity. Following this line of thought, ecological assessments such as the JEF[©], represent a more sensitive tool when assessing more comprehensively the psychopathological spectrum of patients with schizophrenia. Additionally, previous studies have shown concurrent validity of the JEF[©] with other gold standard tests of EF, as well as its capacity to discriminate between healthy controls and patients with mood disorders [22], frontal lobe lesion [21], acquired brain injury [20], cannabis [31] and, even caffein consumption [32]. The JEF[©] has thus gained evidentiary value as an ecological test that can be useful for assessing EF in diverse populations.

Regarding functional outcome, the FAST correlated moderately with both JEF (r = -.567, p < 0.01) and CEF (r = -.474, p < 0.05) and strongly with AES-C (r = .817, p < 0.01). These results are consistent with previous research suggesting that motivational deficits are key predictors of daily life functioning in schizophrenia patients [14, 15, 33, 34] and therefore constitute an important treatment target to improve quality of life in schizophrenia patients.

Regarding the second aim, the mediation analysis revealed a strong association between apathy and EF on the one hand, and apathy and functional outcome on the other hand; however, apathy did not act as a mediator between EF and functional outcome. Although the mediation pathway was not confirmed, we did find that AES-C explained a significantly higher portion of variance of FAST (36.7%) than JEF[©] (28.4%), which suggests that motivational deficits exert a greater impact on functional outcome than cognition. In agreement with previous findings [5, 6, 30], the model found that cognitive impairment does not directly affect functional outcome but rather has an impact on other symptomatic components such as motivation that may ultimately lead to a deterioration of the daily life functioning of schizophrenia patients.

There are some limitations to this study, mainly pertaining to the difficulty in collecting the entire sample, which resulted in a relatively small sample size, as well as the fact that most participants were male and only 3 women were able to finish the protocol. However, we were able to find meaningful and predicted associations with the AES-C, which lends credence to our findings. As another limitation, the JEF[©] is a relatively new ecological test that still needs more validation studies, and in addition, it is a long task that can be tiring for patients with schizophrenia. Additionally, the traditional tests that formed the CEF were not selected to match the same subcomponents of EF that are measured by the JEF, it is therefore suggested that future research should take this into consideration. Lastly, our findings should be replicated with a larger sample to provide more information about cognition and motivational deficits in schizophrenia.

To our knowledge, this is the first study that explores the differences between traditionally used EF tasks and ecological cognitive tasks and their impact on the assessment of the entire cognitive and negative psychopathology of schizophrenia patients. This study also addressed the mediating effect of apathy on cognition and functional outcome using a virtual reality

cognitive task. This is relevant since apathy is characterized by goal-directed behavior and ecological assessments can be a better way to understand its impact on schizophrenia patients.

## Supporting information

**S1 Table. Details of the measures used for cognition, negative symptoms, depression, and functionality.**
(PDF)

**S2 Table. Correlations between cognition, apathy, functionality, clinical variables, and sociodemographic characteristics.**
(PDF)

## Acknowledgments

Daniela Ramos-Mastache is a Ph.D. student from the Programa de Maestría y Doctorado en Psicología de la Universidad Nacional Autónoma de México (UNAM) and received a grant from Consejo Nacional de Ciencia y Tecnología CONACYT, CVU 621340. The authors thank Alan Ricardo Anaya Huitrón for his support in recruiting the study sample.

## Author Contributions

**Conceptualization:** Alejandra Mondragón-Maya, Stefanie Enriquez-Geppert, David Pérez-Ferrara, Andre Aleman.

**Data curation:** Daniela Ramos-Mastache, David Pérez-Ferrara.

**Formal analysis:** Daniela Ramos-Mastache, Edith J. Liemburg, Katharina S. Goerlich, Andre Aleman.

**Investigation:** Daniela Ramos-Mastache, Alejandra Mondragón-Maya, Mauricio Rosel-Vales, David Pérez-Ferrara.

**Methodology:** Daniela Ramos-Mastache, Alejandra Mondragón-Maya, Edith J. Liemburg, Katharina S. Goerlich, Andre Aleman.

**Project administration:** Alejandra Mondragón-Maya, Mauricio Rosel-Vales, Andre Aleman.

**Resources:** Alejandra Mondragón-Maya, Mauricio Rosel-Vales, Andre Aleman.

**Supervision:** Alejandra Mondragón-Maya, Stefanie Enriquez-Geppert, Katharina S. Goerlich, Mauricio Rosel-Vales, Ashok S. Jansari, Andre Aleman.

**Writing – original draft:** Daniela Ramos-Mastache, Alejandra Mondragón-Maya, Andre Aleman.

**Writing – review & editing:** Daniela Ramos-Mastache, Alejandra Mondragón-Maya, Edith J. Liemburg, Stefanie Enriquez-Geppert, Katharina S. Goerlich, Mauricio Rosel-Vales, David Pérez-Ferrara, Ashok S. Jansari, Andre Aleman.

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
