## [Decision Letter · Decision Letter 0]

22 Jul 2022

PONE-D-21-39855

Understanding the relationship between apathy, cognition, and functional status in schizophrenia: The significance of an ecological assessment

PLOS ONE

Dear Dr. Mondragón-Maya,

Thank you for submitting your manuscript to PLOS ONE. After careful consideration, we feel that it has merit but does not fully meet PLOS ONE’s publication criteria as it currently stands. Therefore, we invite you to submit a revised version of the manuscript that addresses the points raised during the review process.

Please note that we have only been able to secure a single reviewer to assess your manuscript. We are issuing a decision on your manuscript at this point to prevent further delays in the evaluation of your manuscript. Please be aware that the editor who handles your revised manuscript might find it necessary to invite additional reviewers to assess this work once the revised manuscript is submitted. However, we will aim to proceed on the basis of this single review if possible. 

The reviewer has raised a number of concerns about the study regarding the overall data analysis and methodology. The reviewer‘s comments are available below.

We look forward to receiving your revised manuscript.

Kind regards,

Johannes Stortz

Staff Editor

PLOS ONE

Journal Requirements:

Reviewers' comments:

Reviewer's Responses to Questions

**Comments to the Author**

1. Is the manuscript technically sound, and do the data support the conclusions?

Reviewer #1: Yes

2. Has the statistical analysis been performed appropriately and rigorously? 

Reviewer #1: Yes

3. Have the authors made all data underlying the findings in their manuscript fully available?

Reviewer #1: No

4. Is the manuscript presented in an intelligible fashion and written in standard English?

Reviewer #1: Yes

5. Review Comments to the Author

Reviewer #1: The current study examined how an ecological cognitive assessment (the Jansari Executive Function Assessment, JEF) is associated with other variables as between cognition, apathy, and functional status in a sample of patients with schizophrenia. Using The Apathy Evaluation Scale (AES), a neuropsychological battery and the JEF, authors assessed 20 patients with schizophrenia. Data were analysed using a hierarchical multiple regression methodology. Finally, mediation analysis was performed to test the associations between the variables of interest. Authors reported a significant portion of the variance in AES (25%) being explained by the performance in JEF. Additionally, apathy explained 36% of the variance in functional outcome. Unfortunately, AES did not mediate between cognition and functional outcome. Generally speaking, is a well written and interesting paper although some aspects could be in need of some deeper analyses. Otherwise, it could be difficult to interpret some conclusions of the study.

- Did the authors consider that traditional neuropsychological tests included in the study are mirroring the same executive function subcomponents measured by the JEF? How can we interpret results if they are not measuring the same subcomponents?

- Another important issue is the analysis of the potential confounders. Some aspects as medication dose, type of antipsychotic, combination with other medications, sociodemographic characteristics could have an effect on cognition and affective variables. Could the authors try to incorporate the analyses of those confounders in their analyses? If not possible, authors could discuss their potential influence in the discussion as a limitation of the study.

- Authors use multiple regression and correlation coefficients and performed many statistical analyses using a relatively small sample. How did the authors determine the sample size? Did they perform any correction for multiple comparisons?

6. PLOS authors have the option to publish the peer review history of their article (what does this mean?). If published, this will include your full peer review and any attached files.

Reviewer #1: **Yes: **Rafael Penadés

---

## [Author Response · Author response to Decision Letter 0]

19 Sep 2022

Dear Reviewer,

We appreciate your time in reviewing our manuscript entitled "Understanding the relationship between apathy, cognition and functional outcome in schizophrenia: The significance of an ecological assessment” which was submitted for possible publication at Plos One journal. We have incorporated changes that reflect your suggestions. You will find our response highlighted for easier location, also you will find the changes made to the text of the manuscript in bold type letter, and at the bottom of each response accordingly. Here is a point-by-point response to your comments and concerns.

Comment: 

- Do the authors consider that traditional neuropsychological tests included in the study are mirroring the same executive function subcomponents measured by the JEF? How can we interpret results if they are not measuring the same subcomponents?

Response: 

- Traditional neuropsychological tests are designed to assess a single cognitive function in isolation. In contrast, ecological tests such as the JEF, are developed to assess several executive functions in one single task (Parsons T. D. (2015). Virtual Reality for Enhanced Ecological Validity and Experimental Control in the Clinical, Affective and Social Neurosciences. Frontiers in human neuroscience, 9, 660. https://doi.org/10.3389/fnhum.2015.00660). An example of this is the selective-thinking construct of the JEF, in which the participant applies various executive functions such as planning, decision making and even cognitive flexibility to solve one single task. In this sense, the JEF might not mirror the assessment of traditional tests, but it does mirror the executive functions. Although constructed differently, both assessments have shown to be equally valid for the evaluation of executive functioning. In previous studies, the JEF has shown concurrent validity with traditional neuropsychological test global scores and with executive functioning subcomponent scores, it also has been able to differentiate between healthy controls and patients with health conditions such as mood disorders and substance consumption (Hørlyck, L. D., Obenhausen, K., Jansari, A., Ullum, H., & Miskowiak, K. W. (2021). Virtual reality assessment of daily life executive functions in mood disorders: associations with neuropsychological and functional measures. Journal of affective disorders, 280(Pt A), 478–487. https://doi.org/10.1016/j.jad.2020.11.084; Soar, K., Chapman, E., Lavan, N., Jansari, A. S., & Turner, J. J. (2016). Investigating the effects of caffeine on executive functions using traditional Stroop and a new ecologically-valid virtual reality task, the Jansari assessment of Executive Functions (JEF(©)). Appetite, 105, 156–163. https://doi.org/10.1016/j.appet.2016.05.021). In this sense, our results can be comparable with traditional neuropsychological assessments, providing a more in-depth type of evaluation since the participants not only rely on their cognitive abilities, but also on their emotional and social resources that can help or interfere with the completion of a tasks. To make this point clearer, we extended the discussion in the revised manuscript by adding information to understand the relationship and validity between the two types of neuropsychological assessments (ecological and traditional) (page 15 lines 254 - 260):

Following this line of thought, ecological assessments such as the JEF©, represents a more sensitive tool when assessing more comprehensively the psychopathological spectrum of patients with schizophrenia. Additionally, previous studies have shown the concurrent validity of the JEF with other gold standard tests of EF, as well as its capacity to discriminate between healthy controls and patients with mood disorders[22], frontal lobe lesion[21], acquired brain injury[20], cannabis[31] and, even caffein consumption[32]. The JEF© has thus gained evidentiary value as an ecological test that can be useful for assessing EF in diverse populations. 

Comment: 

- Another important issue is the analysis of the potential confounders. Some aspects as medication dose, type of antipsychotic, combination with other medications, sociodemographic characteristics could have an effect on cognition and affective variables. Could the authors try to incorporate the analyses of those confounders in their analyses? If not possible, authors could discuss their potential influence in the discussion as a limitation of the study. 

Response: 

- We appreciate your remarks on this point as they improve the quality of our manuscript. Type of antipsychotic and other medications were coded as dichotomic nominal variables, therefore the information provided by these variables were limited to only “yes” and “no” answers. Spearman correlations were conducted between clinical, cognitive, and sociodemographic variables (gender and type of medication) to search for possible confounders. The results showed that benzodiazepines (r = .481, p = .032) and antidepressants (r = .490, p = .028), correlated with CDSS (depression) and antipsychotic medication with BNSS (negative symptoms; r = .509, p = .022). We briefly mention this now in the Results section (page 10, lines 186 - 191). Of note, these correlations were not related to our main variables of interest (AES-C, FAST, JEF and CEF), so we decided to no report them further in the study: 

Results of Pearson correlations between cognitive measures, clinical measures and sociodemographic showed that years of education correlated with AES-C (r = -.661, p < 0.01), JEF© (r = .683, p < 0.01), FAST (r = -.624, p < 0.01), BNSS (r = .800, p < 0.01) and PANSS (r = -.606, p < 0.01) (S2 Table). Further, Spearman correlations between apathy, functionality, cognitive measures and sociodemographic showed that benzodiazepines (r = .481, p = .032) and antidepressants (r = .490, p = .028), correlated with CDSS and antipsychotic medication with BNSS (negative symptoms; r = .509, p = .022).

- Regarding continuous sociodemographic variables (age, years of education and age of onset of illness), bivariate Pearson correlations were calculated. Results showed that years of education correlated with AES-C (r = -.661, p < 0.01), JEF (r = .683, p < 0.01), FAST (r= -.624, p < 0.01), BNSS (r = .800, p < 0.01) and PANSS (r = -.606, p < 0.01). Because of collinearity, no additional analyses were conducted. Nevertheless, a supplementary table (S2 Table) will be uploaded with this information: 

S2 Table. Correlations between cognition, apathy, functionality, clinical variables, and sociodemographic characteristics. 

 1 2 3 4 5 6 7 8 9 10

1. AES-C 1.000 

2. JEF© -.546* 1.000 

3. CEF -.380 .457* 1.000 

4. FAST .817** -.567** -.474* 1.000 

5. BNSS .680** -.636** -.278 .503* 1.000 

6. CDS .083 -.088 .008 .287 .037 1.000 

7. PANSS .423 -.698** -.440 .506* .565** .159 1.000 

8. Age in years -.172 -.160 -.415 .128 -.456* -.049 .172 1.000 

9. Years of education -.661** .683** .410 -.624** -.800** -.298 -.606** .217 1.000 

10. Age of onset of illness -.158 .030 .097 -.161 -.228 .093 -.042 .329 .268 1.000

Note. AES-C=Apathy Evaluation Scale - Clinical version; JEF©=Jansari Assessment of Executive Functions; CEF=Composite score of executive function assessment; FAST=Functionality Assessment Short Test; BNSS= Brief Negative Symptom Scale; CDS= Calgary Depression Scale; PANSS=Positive and Negative Syndrome Scale. 

* significance level at .05, ** significance level at .01. 

Comment: 

- Authors use multiple regression and correlation coefficients and performed many statistical analyses using a relatively small sample. How the authors determined the sample size? Did they perform any correction for multiple comparisons?

Response: 

- Thank you for your comment as it help us make our research method clearer. We are aware that our sample size is small, but we consider that these preliminary results are of interest for further exploration of ecological testing in clinical samples such as schizophrenia and should be published. We did calculate the sample size, initially we were expecting to assess at least 63 schizophrenia patients. Unfortunately, most of the patients withdrew their participation or were unable to complete their assessments due to COVID outbreak. On the Method section we clarify this point describing in more detail the sample size collection, we also add the small sample size as part of the study limitations (page 6 lines 100 - 103; pages 15-16 lines 273 - 276): 

Initially, a total of 63 participants were to be recruited for the study, however due to Covid-related restrictions, only 35 participants could be included. Of these, 11 participants (8 women and 3 men) withdrew their participation after the sociodemographic data collection and four patients (three women, one man) could not complete de assessment. 

There are some limitations to this study, mainly pertaining to the difficulty in collecting the entire sample, which resulted in a relatively small sample size, as well as the fact that most participants were male and only 3 women were able to finish the protocol. However, we were able to find meaningful and predicted associations with the AES-C, which lends credence to our findings.

- Regarding the correction for multiple comparisons, we consider our data meet the following criteria and therefore p value corrections were not required: 

1. The correlations in our study worked as a first step to meet the requirements needed to conduct the regression analysis (moderate correlation between the variables of interest). Therefore, the correlations in our study were meant to be exploratory rather than explicatory and corrections are meant mainly in confirmatory analysis (Cao, J., & Zhang, S. (2014). Multiple comparison procedures. JAMA, 312(5), 543–544. https://doi.org/10.1001/jama.2014.9440). p value corrections were too conservative and would have made us fall on type II error. 

2. In regression analysis, p value correction is not conventionally applied. Regarding regression only two predicted variables were established and one regression model, therefore no correction analysis were required (Andrade C. (2019). Multiple Testing and Protection Against a Type 1 (False Positive) Error Using the Bonferroni and Hochberg Corrections. Indian journal of psychological medicine, 41(1), 99–100. https://doi.org/10.4103/IJPSYM.IJPSYM_499_18).

---

## [Editor Report · Decision Letter 1]

19 Oct 2022

Understanding the relationship between apathy, cognition, and functional status in schizophrenia: The significance of an ecological assessment

PONE-D-21-39855R1

Dear Dr. Mondragón-Maya,

We’re pleased to inform you that your manuscript has been judged scientifically suitable for publication and will be formally accepted for publication once it meets all outstanding technical requirements.

Kind regards,

Rafael Penades

Guest Editor

PLOS ONE
---

## [Editor Report · Acceptance letter]

25 Oct 2022

PONE-D-21-39855R1 

Understanding the relationship between apathy, cognition and functional outcome in schizophrenia: The significance of an ecological assessment. 

Dear Dr. Mondragón-Maya:

I'm pleased to inform you that your manuscript has been deemed suitable for publication in PLOS ONE. Congratulations! Your manuscript is now with our production department. 

Kind regards, 

on behalf of

Dr. Rafael Penades 

Guest Editor

PLOS ONE